# A Novel Positive-Contrast Magnetic Resonance Imaging Line Marker for High-Dose-Rate (HDR) MRI-Assisted Radiosurgery (MARS)

**DOI:** 10.3390/cancers16101922

**Published:** 2024-05-18

**Authors:** Li Wang, Yao Ding, Teresa L. Bruno, R. Jason Stafford, Eric Lin, Tharakeswara K. Bathala, Jeremiah W. Sanders, Matthew S. Ning, Jingfei Ma, Ann H. Klopp, Aradhana Venkatesan, Jihong Wang, Karen S. Martirosyan, Steven J. Frank

**Affiliations:** 1Department of Experimental Radiation Oncology, The University of Texas MD Anderson Cancer Center, Houston, TX 77030, USA; lwang6@mdanderson.org (L.W.); erli2000@gmail.com (E.L.); 2Department of Radiation Physics, The University of Texas MD Anderson Cancer Center, Houston, TX 77030, USA; yding1@mdanderson.org (Y.D.); jihong.wang@mdanderson.org (J.W.); 3Department of Radiation Oncology, The University of Texas MD Anderson Cancer Center, Houston, TX 77030, USA; tbruno@mdanderson.org (T.L.B.); msning@mdanderson.org (M.S.N.); aklopp@mdanderson.org (A.H.K.); 4Department of Imaging Physics, The University of Texas MD Anderson Cancer Center, Houston, TX 77030, USA; jstafford@mdanderson.org (R.J.S.); jma@mdanderson.org (J.M.); 5Department of Diagnostic Radiology, The University of Texas MD Anderson Cancer Center, Houston, TX 77030, USA; tkbathala@mdanderson.org (T.K.B.); avenkatesan@mdanderson.org (A.V.); 6Department of Radiology, Mayo Clinic, Phoenix, AZ 85054, USA; sanders.jeremiah@mayo.edu; 7Department of Physics, The University of Texas Rio Grande Valley, Brownsville, TX 78500, USA; karen.martirosyan@utrgv.edu

**Keywords:** MRI-assisted radiosurgery, brachytherapy, interventional radiotherapy, image guidance, magnetic resonance imaging, positive-contrast marker, high dose rate

## Abstract

**Simple Summary:**

High-dose-rate (HDR) MRI-Assisted Radiosurgery (MARS) is an advanced form of brachytherapy or interventional radiotherapy involving the use of a remote after-loading system to transport an encapsulated radioisotope source to within or near a tumor with MRI precision. An HDR MARS can be used for treating both primary and recurrent prostate cancer with fewer treatment-related toxicities and comparable survival rates. Image-guided HDR brachytherapy has been linked with better local tumor control and overall survival. Magnetic resonance imaging (MRI) is more effective than computed tomography (CT) and ultrasonography in distinguishing the boundaries of a tumor versus the surrounding normal tissues. However, localizing the applicator and source is more challenging with MRI because conventional fiducial markers are not positively visualized on MRI. The aim of our study was to investigate the MRI signal intensities and characteristics of a newly developed positive-contrast MRI line marker (C4:S) for HDR MARS. We found that C4:S MRI line markers were positively visualized on both T1- and T2-weighted MRI sequences. Our findings indicate the value of C4:S line markers for HDR MARS.

**Abstract:**

Magnetic resonance imaging (MRI) can facilitate accurate organ delineation and optimal dose distributions in high-dose-rate (HDR) MRI-Assisted Radiosurgery (MARS). Its use for this purpose has been limited by the lack of positive-contrast MRI markers that can clearly delineate the lumen of the HDR applicator and precisely show the path of the HDR source on T1- and T2-weighted MRI sequences. We investigated a novel MRI positive-contrast HDR brachytherapy or interventional radiotherapy line marker, C4:S, consisting of C4 (visible on T1-weighted images) complexed with saline. Longitudinal relaxation time (T1) and transverse relaxation time (T2) for C4:S were measured on a 1.5 T MRI scanner. High-density polyethylene (HDPE) tubing filled with C4:S as an HDR brachytherapy line marker was tested for visibility on T1- and T2-weighted MRI sequences in a tissue-equivalent female ultrasound training pelvis phantom. Relaxivity measurements indicated that C4:S solution had good T1-weighted contrast (relative to oil [fat] signal intensity) and good T2-weighted contrast (relative to water signal intensity) at both room temperature (relaxivity ratio > 1; r2/r1 = 1.43) and body temperature (relaxivity ratio > 1; r2/r1 = 1.38). These measurements were verified by the positive visualization of the C4:S (C4/saline 50:50) HDPE tube HDR brachytherapy line marker on both T1- and T2-weighted MRI sequences. Orientation did not affect the relaxivity of the C4:S contrast solution. C4:S encapsulated in HDPE tubing can be visualized as a positive line marker on both T1- and T2-weighted MRI sequences. MRI-guided HDR planning may be possible with these novel line markers for HDR MARS for several types of cancer.

## 1. Introduction

The application of high-dose-rate (HDR) MRI-Assisted Radiosurgery (MARS) involves the use of a remote after-loading system to precisely transport an encapsulated radioisotope source to within or near an MRI-delineated tumor, with the goal of delivering high, well-conformed radiation doses to tumors while minimizing the exposure of surrounding normal tissues [1]. The radiation dose is determined from calculations of the source position and dwell time. With modern-day MRI image guidance, HDR MARS can be used as primary, adjunct, or salvage treatment for cancer of the prostate, cervix, breast, and other organs [2,3,4,5,6]. HDR brachytherapy or interventional radiotherapy is an attractive choice for treating both primary and recurrent prostate cancer, either as monotherapy for low-risk or, in some cases, intermediate-risk disease, or as dose escalation with external-beam radiation therapy (EBRT) for high-risk disease, with good quality of life outcomes [2,7,8,9,10,11,12,13,14]. Compared with radical prostatectomy, HDR brachytherapy for prostate cancer leads to fewer instances of treatment-related impotence and urinary disorders as well as comparable 10-year survival rates [2,15]. HDR brachytherapy with EBRT has also led to improved survival rates in cervical cancer [16,17,18,19,20,21,22], and HDR brachytherapy has produced favorable outcomes with fewer treatment-related side effects than traditional EBRT after lumpectomy for breast cancer [2,23,24,25].

In HDR MARS, the radiation source is placed either within or close to the MRI-defined tumor being treated, and high radiation doses are delivered over short periods. Therefore, any errors in source positioning can have significant negative effects on both tumor control (from inadequate dose delivery) and toxicity to surrounding normal tissues (from high-dose exposure), highlighting the importance of the accuracy of both the placement of the implant and the ability to visualize the tumor boundary. HDR brachytherapy has a somewhat higher risk than other radiation modalities of errors related to poor image quality, particularly in catheter reconstruction or catheter/indexer length [7,26,27,28,29]. Indeed, mislabeled catheters and incorrect catheter reconstruction are two primary sources of inaccuracy in HDR [7,28,30]. Accurate delineation of the anatomy, tumor, and implanted catheters or applicators (and therefore safe and precise dose delivery) has relied on image guidance by computed tomography (CT), transrectal ultrasonography, or magnetic resonance imaging (MRI) [1,31,32,33]. Image-guided HDR brachytherapy has been linked with better local tumor control and overall survival [2,33,34,35,36]. Ultrasonography can be problematic because of poor visualization of the catheters or applicators, especially toward the catheter tips [7]. CT is often used for image-guided HDR radiotherapy because it can visualize the applicators, sources, and source path [37,38].

However, CT is less effective than MRI for distinguishing anatomic structures, especially soft tissues, which limits the accuracy of CT for visualizing the boundaries of the tumor versus the surrounding normal tissues. As a consequence, tumor delineations based on CT can significantly overestimate the tumor width relative to MRI [3,39]. Moreover, interobserver variability is less of an issue with MRI than with CT [40]. 

The high soft-tissue contrast possible with MRI has led to it being recommended as a gold-standard modality for visualizing tumor volumes [41,42,43,44], especially cervical and prostate cancers [3,37,45,46]. MRI was also recently named a preferred imaging modality for image-guided HDR brachytherapy, especially for gynecologic and prostate cancer [3,33,46,47], because MRI can guide interventional procedures (including catheter insertion) and assist in detecting interventional devices [2,26,31,48,49,50,51]. Several studies have shown that MRI-guided HDR brachytherapy leads to better cervical cancer-related survival and treatment-related toxicity relative to conventional or CT-guided HDR brachytherapy [2,21,52,53,54,55,56,57,58,59]. 

Although MRI is superior to CT for target volume delineation at several anatomic sites [3,33,40], localizing the applicator and source is more challenging with MRI than with CT [33,37,38], leading to the inaccurate reconstruction of the source pathways and subsequent dosimetric uncertainties in the tumor and surrounding normal tissues [37,60]. Because conventional fiducial markers do not appear as positive contrast on MRI, positive-contrast MRI markers are being developed to improve the accuracy of applicator and source localization and thus improve the precision of dose distribution in HDR MARS [61,62,63]. 

Markers such as copper sulfate (CuSO_4_) and C4 (cobalt chloride complexed with water and N-acetylcysteine) can be positively visualized on T1-weighted MRI, and saline can be visualized on T2-weighted MRI; both have shown promise for identifying source pathways [46]. Because both T1-weighted and T2-weighted MRI sequences are used for HDR MARS in cervical and prostate cancer, markers that appear as positive contrast on both T1- and T2- weighted MRI sequences would be ideal. To meet this need, we investigated the magnetic relaxation characteristics and MRI signal intensities, on both T1- and T2-weighted images, of a newly developed positive-contrast MRI line marker for HDR, in which C4 is complexed with saline (C4:S). 

## 2. Materials and Methods 

### 2.1. Preparation of Test Solutions

The MRI marker C4, which shows a positive signal on T1-weighted sequences, consists of cobalt chloride complexed with 6 water molecules (CoCl_2_·6H_2_O, Fluka Sigma-Aldrich 60820, Lot 1313139, St. Louis, MO, USA) and the chelating antioxidant N-acetylcysteine (NAC, Sigma-Aldrich 47250-1006, Lot 051M1820V), prepared as described previously [64]. Briefly, the ingredients were dissolved in distilled water in a weight-by-volume percentage (*w*/*v*%, where 1% = 1 g/100 mL) of C4, that is, 1% CoCl_2_·6H_2_O: 2% NAC. Saline was created by dissolving 9 g of sodium chloride (NaCl) in 1000 mL of distilled water. The C4 was complexed with saline to create the positive-signal marker C4:S (under both T1- and T2-weighted sequences). The C4:S solutions tested were generated from the following proportions of C4 and saline: 10%:90%; 20%:80%; 30%:70%; 40%:60%; 50%:50%; 60%:40%; 70%:30%; 80%:20%; and 90%:10%. Because the standard units for relaxivity calculations are millimolar concentrations in solution (mM^−1^) and time (in seconds^−1^), the above *w*/*v*% concentrations of the tested solutions were converted to the corresponding mM concentrations of cobalt dichloride, i.e., 4.229 mM, 8.458 mM, 12.687 mM, 16.916 mM, 21.145 mM, 25.374 mM, 29.603 mM, 33.832 mM, and 38.061 mM. 

### 2.2. MRI Scan Acquisition Parameters and Image Analysis

MRI scans were performed on a 1.5 T scanner (Siemens Healthineers, Erlangen, Germany) and images were analyzed as described previously [64]. Briefly, the C4:S solutions (total volume of 25 mL each) were placed in nine 50 mL polypropylene centrifuge tubes (inner diameter was 26.6 mm; thickness was 1.06 mm). Three additional tubes, two filled only with water and the other filled only with oil, were used as controls. The 12 tubes were placed in an acrylic box, as shown in Figure 1. The box was then placed in a 20-channel phased-array head-and-neck coil and centered about the isocenter of the bore, with the length of the vials perpendicular with the static B0 field, and filled with water to reduce susceptibility artifacts. The relaxation parameters of the C4:S solutions were determined both at body temperature (37 °C) and at room temperature (20 °C) in three conventional scan orientations (coronal, sagittal, and axial). MATLAB v9.3.0. (The MathWorks, Inc., Natick, MA, USA) was used to analyze the acquired images. The area of each image analyzed was a 20-pixel-by-20-pixel square region of interest (ROI), which was placed away from the edge to ensure signal homogeneity within the ROI. The mean and standard deviation of the signal within the ROIs were recorded at each time point.

### 2.3. Spin–Lattice (Longitudinal) Relaxation Time (T1) Measurements

For the T1 measurements, an inversion recovery turbo spin echo sequence was used [64], with TI = 50, 100, 200, 400, 600, 800, 1200, 1600, and 2400 ms; matrix size = 192 × 192; FOV = 20 cm; TR/TE = 8000 ms/7.6 ms; pixel bandwidth = 285 Hz; NEX = 1; ETL = 11; and slice thickness = 5 mm. The signal for inversion recovery is expressed as $S=K(1-2e^{-\frac{TI}{T1}}+e^{-\frac{TR}{T1}})$, where *K* is a scaling factor, *T*1 denotes the inversion time, and *TR* is the repetition time. For an inversion recovery scan at a specific C4:S concentration, the signal is represented as the mean signal in the ROI, with standard deviation used to estimate uncertainty. In our experiments, *TR* and *K* were constants. The curve was fitted to the inversion recovery measurements, and the Levenberg–Marquardt least-squares algorithm [65,66] was used to determine *T*1.

### 2.4. Spin–Spin (Transverse) Relaxation Time (T2) Measurement

For T2 measurements, two turbo spin echo sequences were used. The sequence used to investigate the influence of temperature on T2 relaxation time was based on traditional Cartesian sampling, with TR = 5000 ms; TE = 10, 30, 50, 70, 90, 110, 130, 150, 170, 190, 210, 230, 250, 270, 290, and 310 ms; matrix size = 192 × 192; pixel bandwidth = 170 Hz; ETL = 19; FOV = 20 cm; and slice thickness = 5 mm. To investigate the effect of scanning orientation on T2 relaxation time, the sequence used was a fast radial sampling, with TR = 3610 ms; TE = 22, 44, 66, 88, 110, 132, 154, 176, 198, 220, 242, 264, 286, 308, 330, and 352 ms; matrix size = 384 × 384; pixel bandwidth = 130 Hz; ETL = 16; FOV = 20 cm; and slice thickness/gap = 5/5 mm. In the current study, the measured echo signal intensity values were fit to $S=Ke^{-\frac{TE}{T2}}$. The initial estimate of T2 for each ROI was obtained by first-degree polynomial fitting of the turbo spin echo logarithmic signals plotted against echo time. 

### 2.5. Relaxivity Calculation

Relaxivity was calculated as described previously [64]. Briefly, using the initial detected T1 or T2 as a starting point, nonlinear regression was applied to the signal plotted against each inversion/echo time, and the least-squares method was iteratively applied to evaluate T1 and T2. Relaxivity was defined as the change in relaxation rate per unit of C4:S; the relaxation rates were calculated and plotted against the cobalt dichloride concentration. As a result, the slopes from the linear fit to this plot (r2/r1) indicated the relaxivity values.

### 2.6. C4:S Marker Implantation and MRI Scanning

To test visualization of the C4:S marker on MRI, we filled an HDPE tube with C4:S to create a “line marker” that was then implanted in a tissue-equivalent female ultrasound training pelvis (Model 404A) phantom (Figure 2). Image data were acquired with an 18-channel phase-array body coil along with a built-in 32-channel spine coil. MRI scanning was performed with 3D T1-weighted imaging (Fast Low Angle SHot [FLASH]: TR/TE = 6/2.4 ms; NEX = 3; pixel bandwidth = 500 Hz; flip angle = 25 deg; FOV = 18 cm; matrix size = 256 × 256; reconstruction voxel size = 0.35 × 0.35 × 1.25 mm^3^) and 3D T2-weighed imaging (Sampling Perfection with Application optimized Contrasts using different flip angle Evolution [SPACE]: TR/TE = 1500/279 ms; NEX = 2; ETL = 120; pixel bandwidth = 490 Hz; FOV = 18 cm; matrix size = 256 × 256; reconstruction voxel size = 0.35 × 0.35 × 1.25 mm^3^). 

## 3. Results

### 3.1. Relaxation Time and Relaxivity of C4:S at Different Temperatures

Because molecules move differently at different temperatures, and because temperature can influence the characteristics of chemical reactions, our first step was to determine the influence of temperature on the C4:S MRI signal over time. T1 and T2 relaxation times were measured at room temperature (20 °C) and at body temperature (37 °C) in the coronal scanning orientation. At both temperatures, the T1 relaxation time decreased as the proportion of C4 increased; for a 50:50 C4:S solution, the T1 signal intensity was about 119% at room temperature and 116% at body temperature relative to the signal intensity of oil (Figure 3A and Figure 4A,B). Similarly, the T2 relaxation times at both temperatures decreased as the proportion of C4 increased; for a 50:50 C4:S solution, the T2 signal intensity was about 77% at room temperature and 68% at body temperature relative to the signal intensity of water (Figure 3B and Figure 4A,B). Regarding the effects of temperature on C4:S relaxation rates, no difference was observed in either 1/T1 or 1/T2 between room temperature and body temperature when the proportion of C4 in the C4:S solution was low (e.g., the concentration of cobalt dichloride was less than 16.916 mM, which is a less than 40:60 C4:S solution), but when proportions of C4 exceeded 40%, the 1/T1 and 1/T2 values at body temperature were lower than those at room temperature (all *p* < 0.05) (Figure 3C,D). Relaxivity was slightly lower at body temperature (r2/r1 = 1.38) than at room temperature (r2/r1 = 1.43). Moreover, C4:S solutions with a C4:S ratio of 40:60 or 50:50 had the most optimized balance of T1- and T2-relative positive signal intensity, at both room temperature and body temperature (Figure 4).

### 3.2. Relaxation Time of C4:S at Different MRI Scanning Orientations 

Next, to investigate the effects of scanning orientation on C4:S MRI signal artifacts, we measured the T1 and T2 relaxation times of various C4:S ratios in the coronal, axial, and sagittal orientations at room temperature. In all three orientations, the T1 relaxation times became shorter as the proportion of C4 increased, and the times did not differ among orientations (Figure 5A). Similarly, in all three orientations, the T2 relaxation times also became shorter as the proportion of C4 increased, again with no differences noted among orientations (Figure 5B). Similar findings were noted in that orientation did not affect the C4:S relaxation rates (1/T1 and 1/T2), which increased with increasing C4 concentrations (Figure 5C,D).

### 3.3. Positive-Contrast Visualization of C4:S Line Marker under MRI Scanning in a Phantom

As a final test of whether the C4:S marker could be used as a positive-contrast MRI signal marker for HDR, we used a clinical 1.5 T MRI scanner to visualize an HDPE tube filled with C4:S (a “line marker”) that had been implanted in a tissue-equivalent female ultrasound training pelvis phantom. The C4:S (in a 50:50 solution) appeared as a positive signal on both T1- and T2-weighted images, and on T2-weighted images, its visualization was better than C4 (Figure 6).

## 4. Discussion

Our results showed that MRI scanning orientation does not affect the relaxation time or relaxation rate of the C4:S marker. Both the T1 and T2 relaxation times decreased as the proportion of C4 in the C4:S solution increased, both at room temperature and at body temperature, with the relaxation time being slightly higher at body temperature than at room temperature when the C4 concentration was low. Both the T1 and T2 relaxation rates were lower at body temperature than those at room temperature in C4:S solutions with higher C4 concentrations. Moreover, the C4:S 40:60 and 50:50 solutions had the most optimized T1- and T2-relative positive signal intensity, both at both room temperature and at body temperature. Finally, HDPE tubing filled with a 50:50 solution of C4:S was positively visualized on both T1- and T2-weighted MRI sequences, which suggests that C4:S may be an effective positive-contrast MRI marker for visualizing the positions of applicators and catheters in HDR MARS for prostate, cervical, and possibly other types of cancer.

MRI is superior to CT in distinguishing anatomic structures and soft tissues; therefore, the improved MRI visualization of applicator and source pathways is meaningful in HDR brachytherapy clinical practice because it can improve the accuracy and diminish the uncertainties of dose distribution in the tumor and surrounding normal tissues.

The ability of MRI to distinguish anatomic structures depends on the differences in MR signal intensity between the various tissues. The two most common types of MR sequences used in clinical practice are T1-weighted (which often highlights fat tissue) and T2-weighted (which often highlights both fat and water). On an MR image, the signal intensity is directly related to the magnitude of the detected MR signal. The MRI signal intensity of different tissues or materials depends on their magnetic properties, including the T1 (longitudinal) and T2 (transverse) relaxation times. The T1 relaxation time refers to the time constant for regrowth of the longitudinal component of tissue magnetization toward its initial maximum value; tissues or materials with short T1 relaxation times produce high-intensity signals that appear as bright spots on a T1-weighted MR image. On the other hand, the T2 relaxation time represents the time constant for the MRI signal to decay in the transverse plane; tissues or materials with short T2 relaxation times have low-intensity signals and appear as dark regions on T2-weighted MR images [67]. Considering that T1 is always greater than T2, an ideal positive MRI marker that would show hyperintensity on both T1- and T2-weighted images should have a relatively short T1 relaxation time and a relatively long T2 relaxation time.

To generate an MRI marker that is easily visualized on both T1- and T2-weighted sequences, we began with C4, a previously developed marker with positive visibility on T1-weighted sequences [64], complexed with saline solution to create C4:S, which can be visualized on both T1- and T2-weighted sequences. Here, we tested and characterized the relaxation properties of C4:S for this purpose. We found that the relaxation times of both T1 and T2 were decreased as the proportion of C4 increased, which indicated the possibility of selecting optimized tumor and normal tissue differentiation conditions by modulating MRI intensities through adjusting the ratio of C4 versus saline.

The magnetic properties of tissue or materials reflect not only their chemical characteristics but also their physical orientation relative to the primary magnetic field [68,69]; indeed, orientation-dependent brachytherapy seed artifacts have been observed on MR images [70]. Thus, we tested the effect of MRI scanning orientation on the magnetic properties of the C4:S solution, which included the relaxation times and relaxation rates obtained at coronal, sagittal, and axial orientations. Our results indicated that all of the magnetic properties of the C4:S solution were similar in all three orientations, a finding that agrees with our previous studies of the C4 marker [64]. From this result, we anticipate that a C4:S solution encapsulated to mark either a seed or a line (i.e., an applicator) would not induce marker orientation-based artifacts on either T1- or T2-weighted MRI scans, which would help to ensure the accuracy of placement of the radiation seeds and the HDR brachytherapy applicator.

Because T1 and T2 relaxation times are strongly affected by temperature [71,72,73], the T1 and T2 relaxation time values derived from in vitro measurements are not necessarily representative of in vivo conditions. Moreover, the effects of temperature on the relaxation time of an agent also depend on the characteristics of the material [71]. Therefore, we tested the influence of temperature on the magnetic properties of C4:S (relaxation time, relaxation rate, and relaxivity). Both the T1 and T2 relaxation times at body temperature were slightly longer than those at room temperature in a 10:90 C4:S solution, and no differences in the T1 or T2 relaxation times were observed at body temperature versus room temperature in the other C4 concentrations tested. This finding suggests that in vitro tests of C4:S solutions when the C4 component is higher than 10% are applicable to in vivo conditions; it also suggests that a 10:90 C4:S solution may have a higher signal intensity under T1-weighted imaging and a slightly lower signal intensity under T2-weighted imaging than in in vivo conditions. Moreover, both the T1 and T2 relaxivities of C4:S solutions at room temperature were similar to those at body temperature (i.e., the difference was only 0.05 mM^−1^s^−1^), further indicating the feasibility of using in vitro tests to study C4:S MRI properties that may approximate the in vivo conditions well. These results, which are similar to those of studies of the original C4 marker [64], indicate that, unlike some other materials [71,72,73], the experimental data derived from in vitro measurements of C4:S will generally be similar to those present under in vivo conditions.

To further mimic in vivo conditions, we tested whether a 50:50 C4:S solution could be visualized within a tube in a tissue-equivalent phantom of the female pelvis. As expected, the C4:S was positively visualized under clinically used T1- and T2-weighted MRI sequences, and the signal intensity on the T2-weighted image was considerably higher than the intensity of C4. This result indicated the potential of a C4:S line marker to be a clinically applicable MRI positive-contrast marker for HDR interventional radiotherapy guided by MRI using commonly used clinical MRI sequences.

The limitation of this study is that our study was only performed on a 1.5 Tesla MR scanner. Even though the 1.5 Tesla MR scanner is the most widely used field strength for clinical MRI, including MR-Linac systems, 3 Tesla MR scanners are the other field strength widely used for clinical MRI. Although the T1 and T2 of the tissues are field strength-dependent, their changes are fairly small, with T1 increasing by about 25% and T2 becoming only slightly shorter from 1.5 Tesla to 3 Tesla. We expect that the C4:S markers will remain clearly visible on both T1- and T2-weighted images on a 3 Tesla MR scanner, and we will test this hypothesis in our future studies.

## 5. Conclusions

Accurate and precise dose delivery in HDR MARS for cervical or prostate cancer requires the use of both T1- and T2 -weighted MRI sequences [74,75,76,77,78,79]. Therefore, it is essential to be able to visualize the markers under both T1- and T2-weighted sequences. Since a shorter T1 relaxation time usually leads to an increased signal intensity in T1-weighted images, whereas a shorter T2 relaxation time results in a decreased signal intensity in T2-weighted images [67,80], a material with a shorter T1 relaxation time and a longer T2 relaxation time would be an appropriate MRI marker for use in both T1- and T2-weighted sequences. Because the relaxivity ratios of C4:S in our study were greater than 1 at room temperature (r2/r1 = 1.43) and at body temperature (r2/r1 = 1.38), we conclude that C4:S shows promise as a positive-contrast marker for both T1- and T2-weighted MR images and, as such, may be feasible to guide HDR MARS. Further investigations involving the clinical use of C4:S to assist HDR MARS are warranted.

## Figures and Tables

**Figure 1 cancers-16-01922-f001:**
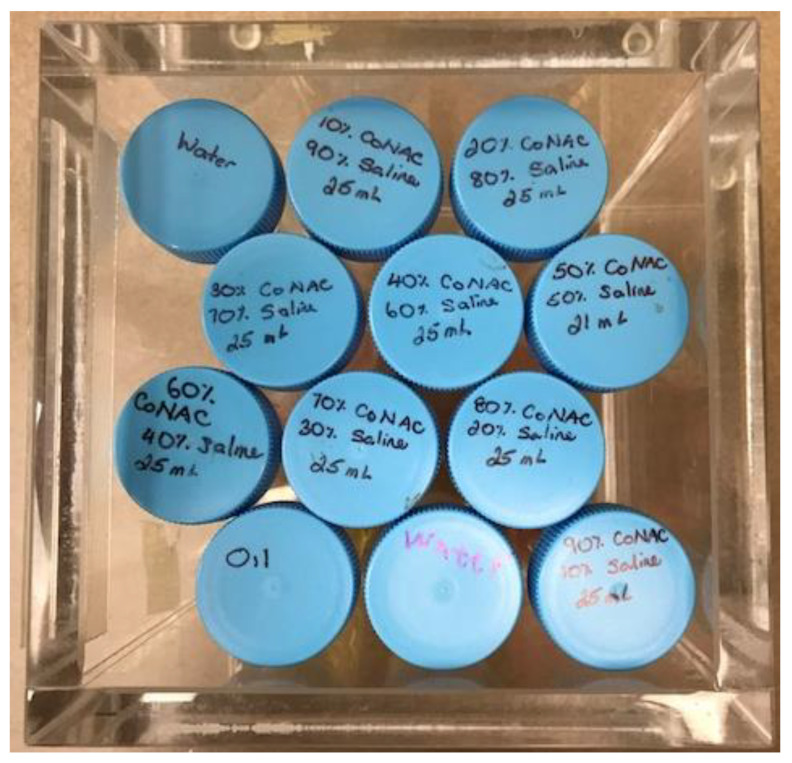
Test solutions for magnetic resonance imaging (MRI). Each of the C4/saline (C4:S) solutions, with C4:S concentrations varying from 10%:90% to 90%:10%, were placed in nine 50 mL polypropylene centrifuge tubes. Three additional tubes, two filled only with water and the other filled only with oil, were used as controls. The 12 tubes were placed as shown in an acrylic box, which was filled with water and placed at the center of the MRI scanning field.

**Figure 2 cancers-16-01922-f002:**
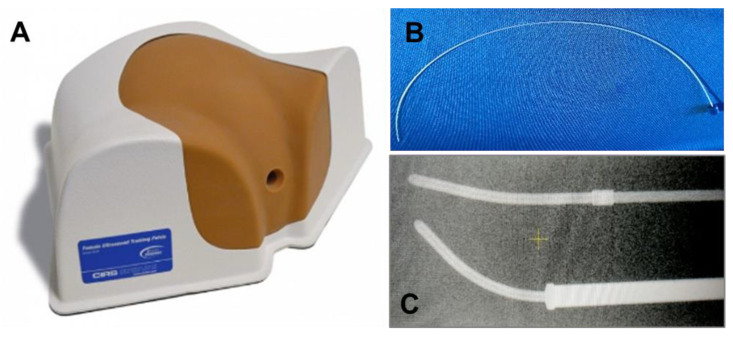
(**A**) Tissue-equivalent female ultrasound training pelvis phantom, model 404A. (**B**) High-density polyethylene (HDPE) tube (thickness: 0.26 mm; outer diameter: 1.0 mm). (**C**) HDPE tubing as visualized on kV imaging.

**Figure 3 cancers-16-01922-f003:**
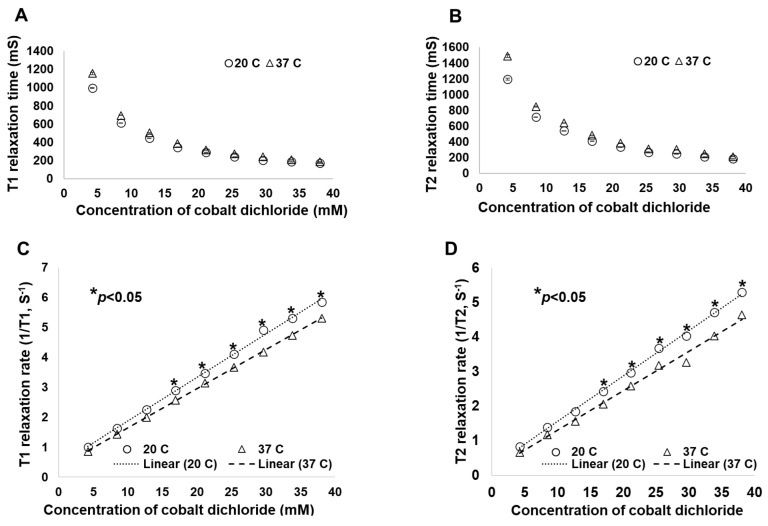
(**A**) Spin–lattice relaxation time (T1), (**B**) spin–spin relaxation time (T2), (**C**) spin–lattice relaxation rate (1/T1), and (**D**) spin–spin relaxation rate (1/T2) at different temperatures for various cobalt dichloride concentrations. Scans were obtained by using a clinical 1.5 T MRI unit in the coronal orientation.

**Figure 4 cancers-16-01922-f004:**
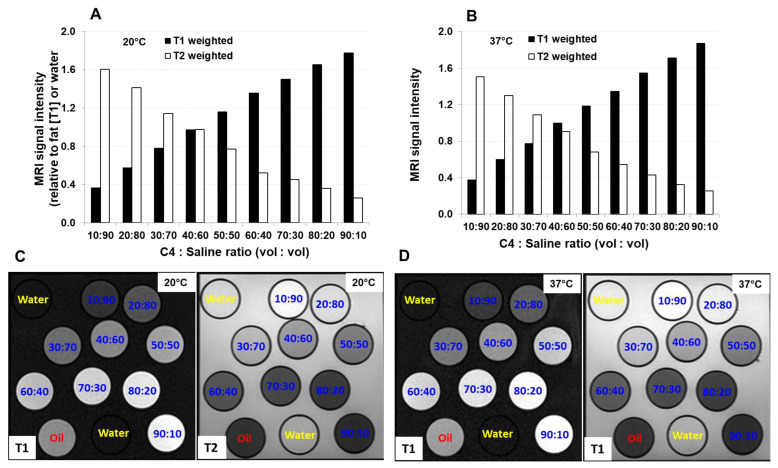
T1- and T2-weighted magnetic resonance imaging (MRI) signal intensities at 20 °C (**A**,**C**) or at 37 °C (**B**,**D**) for various C4/saline ratios. Scans were obtained by using a clinical 1.5 T MRI unit in the coronal orientation.

**Figure 5 cancers-16-01922-f005:**
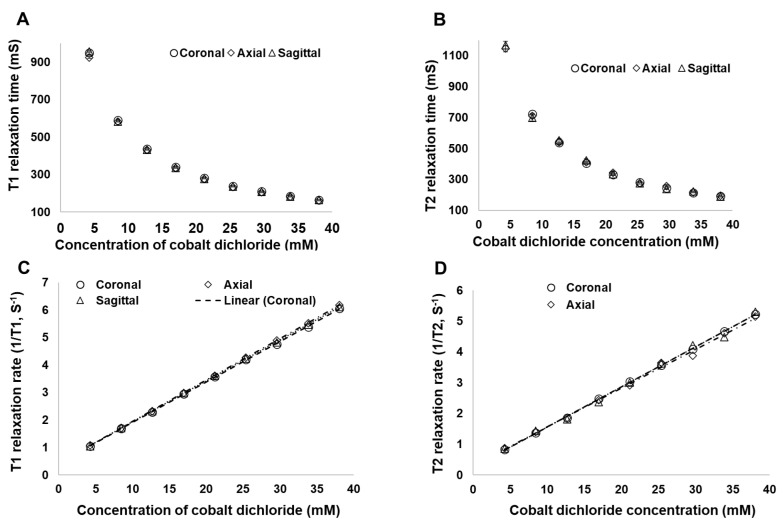
(**A**) Spin–lattice relaxation time (T1), (**B**) spin–spin relaxation time (T2), (**C**) spin–lattice relaxation rate (1/T1), and (**D**) spin–spin relaxation rate (1/T2) as a function of C4 concentration, at room temperature, in different magnetic resonance imaging (MRI) scanning orientations. Scans were obtained by using a clinical 1.5 T MRI unit.

**Figure 6 cancers-16-01922-f006:**
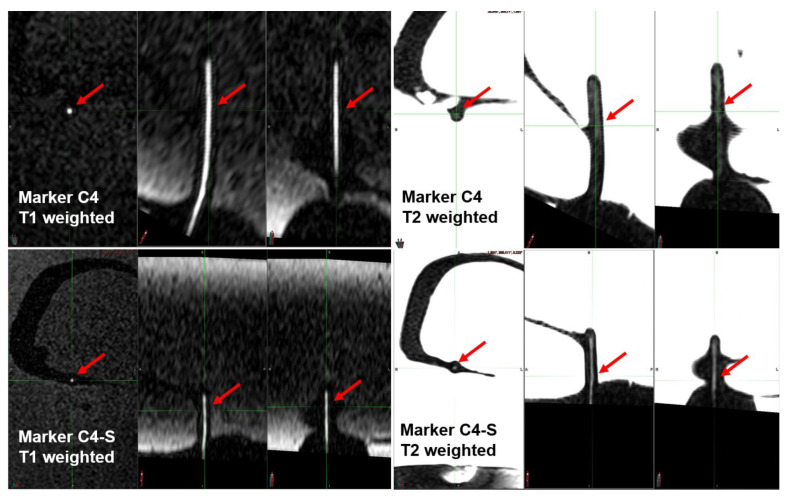
The visualization of C4 and C4:S in MRI images. C4 or C4:S (a 50:50 proportion of C4 and saline) in HDPE tubes implanted in a phantom of a female pelvis were positively visualized in both T1- and T2-weighted sequences; the visualization of C4:S was better than that of C4 in the T2 image.

## Data Availability

The raw data supporting the conclusions of this article will be made available by the authors on request.

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
