# Peer review of "A Novel Positive-Contrast Magnetic Resonance Imaging Line Marker for High-Dose-Rate (HDR) MRI-Assisted Radiosurgery (MARS)"

_cancers, 2024, doi:10.3390/cancers16101922_

Round 1

Reviewer 1 Report

Comments and Suggestions for Authors

The manuscript introduces a very good device which can be used in MRI, for finding the tip and path of the Brachytherapy  applicators. 

Using such markers can be distinguishable in both T1 and T2 weighted MR images can make the clinics needless from using CT Scan for the inserted applicators inside the patients body.

The chemical formula and the concentrations chose and explained in a good manner, also the physics of the MR sequences.

Of course you didn't explain about the used tubes thickness, as you know the diameter of these tubes should not be thicker than 5 Fr. I am not sure if we will have sufficient proton density according to the low volume of the C4:S. Can you explain more about this bug?

I am waiting for your invivo survey, and use the markers in daily practice.

By the way, you can use Interventional Radiotherapy instead of Brachytherapy in your articles.

Author Response

Thank you!

Reviewer 2 Report

Comments and Suggestions for Authors

The authors presented a paper about "A novel positive-contrast magnetic resonance imaging line-marker for high-dose-rate image-guided brachytherapy".

The topic is absolutely interesting because modern interventional radiotherapy (brachytherapy) relies on two major advances: image guidance and intensity modulation.

I would suggest that the authors underline this second concept (intensity modulation) more in detail at least in the discussion because it is strictly linked to the possibility to contour on high quality images.

It would be interesting also to discuss the choice of 1.5 T as reference since there are worldwide several differences (some clinicians use 3 T while others use also hybrid  MRI-Linac of 0.35 T) and to provide (if available) data about other types of scanner.

In addition please specify more in detail the duration of the contrast (since some interventional radiotherapy procedures are delivered twice a day and therefore require a second acquisition at least 6 hours after the first fraction)

Author Response

Thank you!

Reviewer 3 Report

Comments and Suggestions for Authors

The study is well done and the report well-written. I only have a few very minor additional points:

Discussion:

The Limitations section should be added to include concerns raised in Weaknesses which was not mentioned in the study

The Conclusion goes beyond what is warranted by this study. Promising improved diagnostic accuracy is going too far.

The main question addressed by the research is the effectiveness of a newly developed positive-contrast MRI line-marker (C4:S) for HDR brachytherapy. Specifically, the study investigates whether C4:S line-markers can be positively visualized on both T1- and T2-weighted MRI sequences, thereby facilitating accurate MRI-guided HDR planning for various types of cancer. The study is well done and the report well-written. MRI-guided HDR brachytherapy, potentially improving treatment outcomes for various types of cancer. The marker C4:S is designed to address the limitations of conventional markers in accurately delineating the lumen of the HDR applicator and the path of the HDR source on both T1- and T2-weighted MRI sequences.

This study introduces a novel positive-contrast MRI line-marker (C4:S) specifically designed for HDR brachytherapy. While previous research may have focused on MRI markers for various applications, the development of a marker tailored specifically for HDR brachytherapy is a unique contribution.

I only have a few very minor additional points:

Discussion:

Clinical Impact Assessment: Although the study focuses on technical aspects of the marker's visibility, it would be beneficial to include a discussion on the potential clinical impact of improved MRI visualization in HDR brachytherapy.

The Limitations section should be added to include concerns raised in Weaknesses which was not mentioned in the study.

Conclusion:

The Conclusion goes beyond what is warranted by this study. Promising improved diagnostic accuracy is going too far.

Author Response

Thank you!
